# Accurate Indoor-Positioning Model Based on People Effect and Ray-Tracing Propagation

**DOI:** 10.3390/s19245546

**Published:** 2019-12-15

**Authors:** Firdaus Firdaus, Noor Azurati Ahmad, Shamsul Sahibuddin

**Affiliations:** 1Razak Faculty of Technology and Informatics, Universiti Teknologi Malaysia, Kuala Lumpur 54100, Malaysia; shamsul@utm.my; 2Department of Electrical Engineering, Universitas Islam Indonesia, Yogyakarta 55584, Indonesia

**Keywords:** indoor positioning, WLAN fingerprint, people effect, ray-tracing

## Abstract

Wireless local area networks (WLAN)-fingerprinting has been highlighted as the preferred technology for indoor positioning due to its accurate positioning and minimal infrastructure cost. However, its accuracy is highly influenced by obstacles that cause fluctuation in the signal strength. Many researchers have modeled static obstacles such as walls and ceilings, but few studies have modeled the people’s presence effect (PPE), although the human body has a great impact on signal strength. Therefore, PPE must be addressed to obtain accurate positioning results. Previous research has proposed a model to address this issue, but these studies only considered the direct path signal between the transmitter and the receiver whereas multipath effects such as reflection also have a significant influence on indoor signal propagation. This research proposes an accurate indoor-positioning model by considering people’s presence and multipath using ray-tracing, we call it (AIRY). This study proposed two solutions to construct AIRY: an automatic radio map using ray tracing and a constant of people’s effect for the received signal strength indicator (RSSI) adaptation. The proposed model was simulated using MATLAB software and tested at Level 3, Menara Razak, Universiti Teknologi Malaysia. A K-nearest-neighbor (KNN) algorithm was used to define a position. The initial accuracy was 2.04 m, which then reduced to 0.57 m after people’s presence and multipath effects were considered.

## 1. Introduction

Indoor-positioning system (IPS)-based services have great economic potential—Estimated to reach a market value of US$ 10 billion in 2020 [1]. According to a Research and Markets report, the global indoor-positioning and navigation market is expected to reach $54.60 billion by 2026 [2]. IPS utilizes many existing technologies such as radio frequencies (RFs) [3], magnetic fields [4], acoustic signals, and thermal [5], optical [6] or other sensory information collected using a mobile device (MD) [7]. The RF technology used in IPS include WLAN/Wi-Fi [8,9], Bluetooth [10], Zig Bee [11], Radio-frequency identification (RFID) [12], frequency modulation (FM) [13], and ultra-wideband (UWB) [14].

Indoor positioning can be classified into device-based and device-free. On device-based systems, users need a device to know their position, such as smartphone-based and tag-based indoor positioning [15]. Instead of a device-free system, the user does not need a device to know his position. Users here can be people or objects. Device-free localization based on signal strength (received signal strength indicator, RSSI) has three main techniques: fingerprinting, link-based, and radio tomographic imaging. From the user side, device-free is more practical. However, from the system side, it will be more complex, for example it takes 6 to 20 transceiver nodes for fingerprinting techniques, whereas with the same technique device-based only requires 3 transmitters. In fact the number of transmitters (access points, APs) available in a building is limited.

Fingerprinting techniques can be applied to device-based and device-free systems. On device-free systems, a number of nodes (transceivers) will be installed on the building. Then each node will record the RSSI emitted by other nodes and forward the data to the computer for the positioning process. On device-based systems, the device held by the user will record RSSI from several transmitters in the building and can use the same device to determine its position. WLAN fingerprinting can work either based on measurements of the RSSI or of the channel state information (CSI). CSI-based signal fingerprinting provides better accuracy [16]. However among many devices available on the market, CSI is only available on a few devices using modified drivers.

Another thing to consider is that if there are many users whose locations will be detected, the device-free system will find it difficult to identify each user and his position. On the contrary, it is quite easy for device-based. The fact is that almost everyone has a smartphone now, so it is easy to apply device-based localization.

WLAN technology is normally used in IPS because radio waves can pass through obstacles such as floors, walls, ceilings, and human bodies. Hence, a WLAN positioning system could be implemented over a wide coverage area because it does not need any additional devices.

WLAN IPS has been highlighted as a preferred technology indoors due to its accurate positioning results and minimal infrastructure cost [17]. WLAN is a wireless local network standard (IEEE 802.11) that is supported by most mobile phones. However, the WLAN signal is greatly influenced by environmental conditions, which could decrease the accuracy. Examples of obstacles that can cause fluctuations in RSSI are walls, ceilings, and/or people [18,19]. Walls and ceilings have been discussed in depth in past studies [20,21,22]. People’s effect on signal strengths at band waves of 60 GHz, [23], 18–22 GHz, and [24] 2.4 GHz has also been investigated in past studies [25].

The movement of humans in wireless networks is one of the major causes of significant RSSI variation [26,27]. Iyad [28] presented an experiment to show that people’s presence between the access point (AP) and the mobile device (MD) reduced the RSSI by 2 dBm to 5 dBm. This decline in RSSI could result in a position error of more than 2 m. However, Iyad [28] only discussed the effect of one or two people on the RSSI and used a single path signal propagation model to analyze RSSI. Nevertheless, multipath signals such as reflections also have a significant effect on indoor propagation. One of the signal propagation models that have considered the multipath effect is the ray-tracing model. Because this model accounts for the effect of people’s presence and multipath propagation, it is a high-accuracy IPS.

## 2. Related Works

Location-detection techniques in IPS can be categorized into three general categories: Proximity, triangulation, and fingerprinting. WLAN-based RSSI fingerprinting was chosen in this research because this technique can provide highly accurate position estimation at minimum cost. Minimum cost, in this case, indicates that no extra devices are needed to implement the technique. The fingerprinting pattern recognition is done by combining RF with location information. WLAN fingerprinting is done in 2 phases: offline and online.

In the offline phase, a site survey is done to get the vectors of the received signal strength indicator (RSSI) from all the detected access points (APs) at many reference points (RPs) of certain locations. This data forms the radio map (RM) database. In the online phase, a user samples or measures an RSSI vector at his/her position. Then, the system compares the RSSI vector with the RM database. The position is therefore estimated based on the RSSI vector that is most similar to the RM database [19].

### 2.1. Radio Map Construction

Constructing a manual radio map in the offline step is a time-consuming process, especially in large urban areas, as the RSSI value has to be collected at each point in a building [29,30]. Automatic radio map generation was thus developed to reduce the time required for this process [31,32,33]. Du, Yang, and Xiu [32] proposed RSSI geography weighted regression (RGWR) to solve the problem of fingerprint database construction. However, this method requires modified WLAN AP and anchors, so it is less practical and incurs high cost because it requires an additional device. Lin [33] and Ferris [34] proposed an unsupervised simultaneous localization and mapping (SLAM) system for automatic floor map and radio map construction. Lin’s [33] solution, however, only works for room-scale indoor positioning and depends on crowd-sourced data from users to complete missing parts of the floor map and to update the radio map. Ferris [34] proposed a novel technique for solving the Wi-Fi SLAM problem using the Gaussian process latent variable model (GPLVM) to determine the latent-space locations of unlabeled signal strength data. The mean localization error is still quite high 3.97 ± 0.59 m. Then Shen et al. [35] proposed an indoor pathway mapping system that can automatically reconstruct internal pathway maps of buildings without any a-priori knowledge about the building, such as the floor plan or access point locations. The experiments demonstrate that Walkie-Markie is able to reconstruct a high-quality pathway map for a real office-building floor after only 5–6 rounds of walks, with accuracy gradually improving as more user data becomes available. The average and 90 percentile localization errors are 1.65 m and 2.9 m.

A hybrid mechanism that combines manual data collection and user collaboration was proposed by Kim [36] and Suining [37]. Kim’s [36] solution was able to reduce the time and manpower needed to generate the RM database, but the accuracy of the system is strongly influenced by the accuracy of the pedestrian dead reckoning (PDR). Suining proposed a new mechanism, called BCCS (Bayesian Compressive Crowdsensing), by combining crowdsourcing and bayesian compressive sensing to incentivize the signal map construction. BCCS infers the missing values during sparse crowdsensing and reduces the platform sensing cost. However the RSSI error is still quite high 4.5 dB and the localization error is 5 m (Cumulative distribution function (CDF) 80%). Zhuang [38] attempted to improve the performance of the system by implementing a trusted portable navigator (T-PN) to develop a radio map automatically. However, this solution still requires an additional device and returned quite a high error. Gu, Chen, and Zhang [39] further reduced the efforts to collect fingerprint data. Their study reduced the time for collecting the fingerprint, but still yielded quite high errors. The results showed that when sparsity rank singular value decomposition (SRSVD) was used based on 5% of the original data, the error rate was 14%. Jingxue et al. [40] proposed a fast construction method using adaptive path loss model interpolation.

In 2017, Iyad [31] used a multi-wall signal path loss model to generate a radio map automatically. The model, however, had to incorporate knowledge of the environmental layout and AP location information. The model could quickly generate a radio map, but it only considered the direct signal from the transmitter (AP) to the receiver (MD), although indirect signals such as reflection also have a significant influence on indoor signal propagation. This research proposes a propagation model that accounts for multipath effects, namely the ray-tracing model, to obtain a more accurate radio map to improve IPS accuracy.

The matching process uses a positioning algorithm that includes deterministic [41] and probabilistic methods [42]. The probabilistic approach is based on probability methods, such as Naïve-Bayes to compute the probability characteristics of RM instances to find the best-matched fingerprint to determine the current location of MD. On the other hand, the deterministic approach uses scalar values, such as mean or median values, and non-probabilistic methods to determine the current location of the MD. Some examples of deterministic approaches are the K-nearest-neighbor (KNN); the artificial neural network (ANN); and support vector machine (SVM). The deterministic approach is well-established and requires less computational cost since it does not require a large RM as compared to the probabilistic approach.

KNN is a simple and powerful classifier [43]. The algorithm validates the distances in a signal space from the current fingerprint to the fingerprints in the RM database. It also selects the k-nearest positions. The Weighted-KNN (WKNN) technique is used to weigh the influence of each of the closest k-neighbors based on its distance from the query instance [44]. KNN based on the fingerprinting approach is widely used in indoor localization systems due to its high localization accuracy. It is very simple and fast because it compares an online instance with training dataset entries. Therefore, this research was conducted using the deterministic approach and KNN was adopted as the positioning algorithm because of its simplicity and high accuracy, besides requiring minimum computational power and time.

### 2.2. People Presence Effect

Many studies have modeled static obstacles such as walls and ceilings, but few have modeled the effect of people presence. Human bodies absorb, reflect, and diffract WLAN signals, which could affect the value of RSSI. Thus, if offline mapping were performed with no or few people and positioning were performed with many people, the system could lose reliability. The results of past studies have shown that, on average, the presence of human bodies increased the error rate by 11% regardless of the algorithm used [45].

The human activity around MS affects WLAN signal strength [26]. One past study observed a relationship between the fluctuations in RSSI and people activity within a WLAN coverage area. This relationship can be used to build better human detection and tracking systems [27]. Koda et al. [46] measured the attenuation of IEEE 802.11ad WLAN signals in a 60 GHz band caused by human blockage. Subsequently, Slezak et al. [25] conducted empirical studies of human-body blockage in mm-Wave communications.

Booranawong et al. [26] proposed well-known filtering methods (i.e., the moving average and the exponentially weighted moving average filters) and the span thresholding filter to reduce RSSI variations and obtained an estimated position error because of human movement. Unfortunately, the size of the room used for the test was quite small and the error was still above 1 m.

Iyad et al. [28] presented an experiment and found that people’s presence between the AP and the MD reduced the received signal strength by −2 dBm to −5 dBm. Following that, Iyad et al. [47] also performed other experiments to prove that the effect of human presence could be determined based on the distance between the MD and the people, also termed as the influence distance. Both studies explained the effect of people presence around the MD on RSSI and attempted to determine this effect on the quality of IPS. The result was improved positioning accuracy from 1.9 m to 1.7 m. Although the technique succeeded in improving accuracy, it still needs further research to obtain significant accuracy improvements. Iyad [47] subsequently used a multi-wall propagation model that only focuses on signals directly from the transmitter to the receiver. This current research proposes new techniques that consider the multipath effect such as reflection to overcome radio map generation and the issues of the people presence effect (PPE), to obtain a signal accuracy of less than 1 m.

## 3. Materials and Methods

The proposed model, an accurate IPS based on people effect and ray-tracing propagation (AIRY), is outlined in Figure 1. This model was derived from the basic IPS fingerprinting model with some additional features added to overcome the issues explained in the previous section. The first additional feature is an automatic radio map generator using ray-tracing (ARM-RT) to solve the time-consuming problem of manual radio map construction. ARM-RT can provide a fast and accurate radio map. The second additional feature is RSSI adaptation to solve the fluctuation of RSSI due to PPE to enhance the accuracy of the IPS.

In the offline stage, an automatic radio map is created using a ray-tracing propagation model. The input is a map of the building, the AP location (*x*, *y*, *z*), and the transmittance power of each AP. In this study, 3 APs with 2.9 m height were used. There are only 3 APs on the whole floor (80 × 16 m) of level 3 Menara Razak, but the detected Wi-Fi signal on the 3rd floor can be up to 6 SSID that come from other floors. The system was selected 3 APs based on their MAC address. Then, the location estimation was done in the online stage. This process was done by matching the initial RSSI vector measured by the user at a specific position with the RSSI vectors in the RM database [19]. The KNN algorithm was adopted as the fingerprint matching algorithm because of its acceptable accuracy and its relevance with the limitations of mobile devices [48].

The KNN algorithm requires two parameters: (1) “k”, which is the number of the considered nearest neighbors, and (2) the distance function. The distance function was set to the Euclidean distance, while “k” was set to a value of 3 to get the best accuracy [48]. The output of this process is an initial location estimation, which is still less accurate because it has not incorporated the effects of people around the user. Thus, if offline mapping were performed with a few people followed by positioning with many people, the system could lose reliability. To improve accuracy, the next step must incorporate the conditions around the initial position, especially the people around the initial position. Based on the position of people around the MD, an RSSI adaptation process was applied to the RSSI vectors to produce adapted RSSI vectors (ARV). Then, the ARV was matched with the RM, as per the previous process using the KNN algorithm. As a result, accurate position estimations are obtained.

The adopted methodology of this research is shown in Figure 2, consisting of 5 steps, starting from the development of a manual radio map until the validation of the proposed model.

### 3.1. The Development of a Manual Radio Map (MRM)

The first step is the development of a manual radio map (MRM) database. The goal of this step is to build a testing set that will be used to calibrate and validate the proposed solution of this research. This step uses a Wi-Fi scanner Android application to collect information (MAC address and RSSI) from the IEEE 802.11 beacon frames broadcasted by the APs.

Data were recorded at points separated by a distance of 1 m. A total of 30 RSSIs were recorded at each point and the median value taken as the radio map data. Data retrieval was undertaken when there were no people around the MD. The collected RSSI and location data were then stored in a Microsoft Excel file, which was later used for simulation and computing in MATLAB software. The location of the study was Level 3 of Menara Razak, Universiti Teknologi Malaysia Kuala Lumpur (size of 80 m × 16 m).

The RSSI measurement procedure is the mobile device fixed on a tripod in all experiments. The MD height is 1 m above the floor. Then the experimenter walked away about 10 m from the MD when measuring each point. The experimenter pressed the button on the android application used to record RSSI. There is a time delay feature in the application before starting the RSSI recording so that there is time for the experimenter to walk away from the MD. After the RSSI recording is finished there is a beep sound so the experimenter knows when he can return to the MD.

### 3.2. The Development of an Automatic Radio Map Using Ray Tracing (ARM-RT)

The second step involves the development of an automatic radio map using a ray-tracing propagation model (ARM-RT). The details of this device and the process of ARM-RT construction are shown in Figure 3. The inputs are a map of the building, information on material type, relative permittivity, transmitter location (AP) in (*x*, *y*, *z*), the transmit power of AP, and the receiver (MD) height. The output is the RSSI for all rooms, where the RSSI is calculated using a ray-tracing model at every 1 m distance, the same position as that of MRM. Then, the RRSI, which is the output of the modeling, was compared with MRM and an automatic radio map using a multi-wall model (ARM-MW) as per Iyad [48] was constructed using mean squared error (MSE).

The ray-tracing (RT) method was based on ray optics, which solves Maxwell’s equations in the high-frequency regime. Thus, the RT method is a general propagation modeling tool that provides estimates of path loss, the angle of arrival/departure, and time delays [49]. The RT model was initially applied in optical propagation models. The RT model is based on the ray-optic approximation of the propagating field [50] and the uniform theory of diffraction (UTD) [51]. Following that, RT started to be applied in the field of radio propagation in the early 1990s [52]. RT is used for received signal strength (RSS) prediction in urban [53] and indoor environments [54].

The RT technique is an approach that can obtain channel characteristics by identifying the contributions of individual multipath components and then calculating their composition at the receiver, as outlined in Equation (1) [55].
(1)$${\vec{E}}_{r}={\vec{E}}_{0}\left\{\prod {\overset{=}{R}}_{i}\right\}\left\{\prod {\overset{=}{T}}_{i}\right\}\left\{\prod e^{-{j\beta }_{0}l_{i}}\right\}\mathrm{SF}$$

${\bar{E}}_{0}$ is the field on the unit sphere; $\prod {\overset{=}{R}}_{i}$ and $\prod {\overset{=}{T}}_{i}$ are, respectively, the reflection and transmission coefficient dyads along the whole ray path; $\prod e^{-{j\beta }_{o}l_{i}}$ is the product of the propagation phase variations of this ray contribution starting from the unit sphere; and $\mathrm{SF}=\sqrt{A_{0}}/\sqrt{A_{r}}$ is the spreading factor. Additionally, $A_{0}$, and $A_{r}$ are the cross-sectional areas at the unit sphere and the receiver point, respectively [56].

The dielectric properties of the materials affect the field strength of the propagation paths. Permittivity is one of the main material qualities that is important to consider in RT. Permittivity is typically denoted by ε. Permittivity is a measure of how much the molecules in a material oppose the external E-field. Ray tracing uses relative permittivity $\left(\varepsilon_{r}\right)$ or a dielectric constant. The permittivity of a medium is expressed as the product of the dielectric constant and free space permittivity $\left(\varepsilon_{0}\right).$ Each material, including human tissue, has a different relative permittivity value. The relative permittivity of building materials is listed in Table 1.

The building map image file was first changed to csv file format containing the coordinate of each wall, floor, ceiling, as well as relative permittivity value. The server then calculated the RSSI using MATLAB by running the ray-tracing model and storing the ARM-RT database. Relative permittivity values were referred from the databases of previous studies (Table 1). The procedures started with writing the program script based on the ray-tracing propagation model then by defining the layout and parameters of the building; running the simulation; and comparing it to MRM by calculating mean squared error (MSE). The pseudocode of the ray-tracing propagation model is presented in Algorithm 1.
**Algorithm 1:** Automatic Radio Map Ray-Tracing (ARM-RT) Generation
**Input**: Building Map, Relative Permittivity List,AccessPoint List 
**Output**: Automatic Radio Map Ray Tracing(ARMRT)**1:**Defining a Finite Panel (Walls)**2:**3D Formation of the Structure**3:**Calculating Fresnel Coefficients for Walls**4:**For i←1 to size (wall.xyz1,1) do**5:**find reflection and transmission Fresnel coefficient for each wall**6:**End For**7:**  Meshing the Boundary Volume and Assign the RX height **8:**  Calculating the Distance of TX(s) from every mesh node RXi
**9:**For i←1 to  do**10:**find the distance **11:**End For**12:**Equating the Panels (Walls) in 3D**13:**  finding the projection of TX on each panel**14:**  calculating the reflection (mirror) of TX across each panel**15:**Calculating the 2nd Image of TX across each wall**16:**Calculating LOS Components**17:**Calculating Multipath & Reflection Components**18:**find the reflection coefficient**19:**count the walls between reflection paths**20:**find the antenna gain and beam departure angle**21:**calculating the angle of arrival**22:**find the walls between TX and the reflection point**23:**find finite walls between reflection point and RX**24:**find the number of walls between reflection paths**25:**calculate the received signal at RX from the reflection point on wall j **26:**calculate 2nd reflection **27:**Drawing the Maps **28:**LOS Propagation Map Only (a)**29:**1st Reflection Propagation Map Only (b)**30:**2nd Reflection Propagation Map Only (c)**31:**LOS and Propagation Map=a+b+c**32:**Return ARMRT

Information on the building layout and type of material used is required to run ray-tracing simulations. The building layout is given in Figure 4 while the type of material is described in Table 2. Then the parameters of ray-tracing simulation in Matlab is shown in Table 3.

The shape, size, and relative permittivity value of each wall, floor, and ceiling were defined according to the type of material used to construct them. The ray-tracing method is based on ray optics, which solves Maxwell’s equations in the high-frequency regime. Thus, the ray-tracing method is a general propagation modeling tool that provides estimates of path loss, the angle of arrival/departure, and time delays [49]. The ray-tracing technique has been verified in numerous past works as a promising method for indoor radio propagation modeling. The RT model is a good solution to provide an accurate, site-specific field prediction and multidimensional characterization of the radio propagation channel in the time, space, and polarization domains. This is because the model has an intrinsic capability to simulate multipath propagation [58].

The image method was used for the ray-tracing model in the current simulation. RT considers transmission and reflections to predict RSSI. Although using three reflection rays is not an important contributor to signal strength [52]; when considering more than 2 reflections, up to 6, the signal strength changes by only 1 dB, but the complexity and execution time increase exponentially [59]. Therefore, two reflection rays were used in the current simulation. The detail of the parameters used in the simulation is listed in Table 4.

If a ray is reflected/transmitted one or more times before reaching the field point, the ray is called a reflected/transmitted ray. This ray corresponds to the reflection/transmission of EM waves at the interface between different mediums. The propagation direction of a reflected/transmitted ray is determined by the law of reflection/refraction. The magnitude of the reflected/transmitted field is determined by Fresnel’s equations for different polarizations [49]. The procedure is recursive and can be implemented conveniently in a computer program.

### 3.3. The Development of Received Signal Strength Indicator (RSSI) Adaptation

The third step is the development of RSSI adaptation. This step aims to determine the effect of people on RSSI. This step obtains the delta RSSI and takes it as the line of sight (LOS) or non-line of sight (NLOS) PPE constant to use in RSSI adaptation—Explained in detail in the fourth step. Delta RSSI consists of the different values of RSSI when there is one person or more and when there are no people. Three experiments were performed in this step. The purpose of the first experiment is to determine the effect of one person on the RSSI in the LOS position. Meanwhile, the second and third experiments aim to determine the effect of one person and many people on RSSI in the NLOS position. The people in the LOS position are the people that are blocking the direct signals from the AP to te hMD. These experiments were conducted at the auditorium of the Faculty of Industrial Technology, Universitas Islam Indonesia, Yogyakarta, in a room measuring 13 m × 17 m.

The required devices to record RSSI are 3 Access Points (Linksys WAP300N from Linksys and made in Taiwan), 1 Mobile Device (Xiaomi Redmi note 3), and an Android application. The APs (Linksys WAP300N) in this experiment is setting as Wi-Fi Access Point (default) operation mode and 2.4GHz frequency band. There is no beam-forming configuration here. The first procedure for all three experiments involved the setting up of APs and MDs with varied AP heights (1 m, 2 m, 3 m), whereas the height of the MD was set to 1 m. Therefore, the distance between the AP and the MD was 10 m. The MD recorded 30 RSSIs at each position.

In the first experiment, one person would stand between the AP and the MD in the LOS position, with varying distances from the MD (r) ranging from 1 m to 9 m, as shown in Figure 5. In the second experiment, one person would stand around the MD in the NLOS position, with varying distances from the MD (1 m, 2 m, 3 m). In the third experiment, many people would stand around the MD in the NLOS position, with varying distances from the MD (1 m, 2 m, 3 m).

In the second experiment, the position of the people was set up to not directly block the signal from the AP to the MD, i.e., only one person’s position around the MD was used. The people’s position in this experiment was divided into three rings, representing a people position of 1 m, 2 m, or 3 m from the MD. The variations in the people’s positions can be seen in Figure 6 for the first ring, Figure 7 for the second ring, and Figure 8 for the third ring. As per the previous experiment, 3 APs were used in this experiment. The height of the access points was set to 1 m for AP1, 2 m for AP2, and 3 m for AP3, and the height of the mobile device was set to 1 m from the floor.

The third experiment aimed to determine how the presence of many people around MD would affect RSSI. This experiment used 2 to 13 people with a varied position and distance to the MD starting at a distance of 1 m (first ring), as shown in Table 5, followed by a distance of 2 m and 3 m, as described in Table 6, Table 7.

### 3.4. The Development of an Accurate Indoor-Positioning System (IPS) Based on People Effect and Ray-Tracing (AIRY)

The basic concept of the fingerprint technique has been explained in the literature review section. In the offline stage, the automatic radio map database was obtained using ray-tracing propagation. In the online stage, the user measures the RSSI in which he/she stands and holds the MD. Then, the measurement results are referred to as the initial RSSI vectors. These initial RSSI vectors were matched with the ARM-RT database using the KNN algorithm, which produces initial position estimations.

The additional steps in the proposed method involved counting the adapted RSSI vectors to obtain an accurate position. Adapted RSSI vectors were calculated based on the initial RSSI as input and considering the presence of people around the initial position. If people were present around the initial position, then the next step involved detecting whether the people positions were in the LOS or NLOS positions. The LOS position means that the people stood to block the direct path between the transmitter (AP) and the receiver (MD) while the NLOS position meant that the position of the person did not block the direct path.

The following rules were applied for the adaptation of the RSSI vector:If there are people in the NLOS position between the AP position and the initial position, then, the RSSI is added with an “NLOS PPE constant”.If the people are in the LOS position between the AP position and the initial position, then the RSSI is added with a “LOS PPE constant”.Then, if there are no people, the RSSI value does not change.

The value of both parameters, NLOS PPE and LOS PPE constants, was calculated in the previous step. After obtaining an adapted RSSI vector, the last step involved calculating accurate positions using the KNN algorithm. Finally, the accuracy of the accurate position was compared to the initial position.

### 3.5. Validating the Proposed Method

This step consists of two validation processes—the validation of ARM and the validation of AIRY. The validation of ARM was done by calculating the error of the system. To evaluate the positioning method, different performance metrics including accuracy, precision, and responsiveness have been proposed in past studies. Liu et al. [60] used accuracy, complexity, scalability, robustness, and cost as performance metrics. On the other hand, Farid et al. [18] used accuracy, responsiveness, coverage, adaptiveness, scalability, cost, and complexity. Dardari [61] used position estimation error, coverage, robustness, and scalability.

One of the most important features of a localization system is the accuracy of the user/device position [62]. Indoor environments provide a challenging space for localization systems to operate due to the presence of obstacles and multipath effects. Therefore, the system needs to limit the impact of multipath effects and other environmental noises to obtain highly accurate estimates. Positioning accuracy is used as a performance metric in this research. Positioning accuracy refers to the difference between the estimated position and the actual position [63] as in Equation (2):(2)$$\mathrm{Accuracy}=\sqrt{{\left(x_{a}-x_{p}\right)}^{2}+{\left(y_{a}-y_{p}\right)}^{2}}$$

Accuracy can be calculated as the root of the squared difference between each pair of location points where $\left(x_{a},y_{a}\right)$ is the coordinate of an actual point and $\left(x_{p},y_{p}\right)$ is the coordinate of the predicted point.

## 4. Results and Discussion

### 4.1. The Development of a Manual Radio Map

This experiment was carried out on the third floor of Razak Tower, UTM Kuala Lumpur, where RSSI was obtained at 523 points from 20 rooms and corridors. The maximum values of RSSI were −50 dBm, −40 dBm, and −45 dBm for the RSSI transmitted from AP1, AP2, and AP3, respectively. Following that, the minimum RSSI were −89 dBm, −84 dBm, and −87 dBm. The standard deviations for ${\mathrm{RSSI}}_{1},{\mathrm{RSSI}}_{2},{\;\mathrm{and}\;\mathrm{RSSI}}_{3}$ were 7.1, 11.06, and 10.45, respectively. This manual radio map (MRM) was used for the calibration of automatic radio map (ARM) generation.

### 4.2. The Development of an Automatic Radio Map Using Ray-Tracing

ARM is illustrated in Figure 9. There are 3 types of RSSIs: line of sight, first reflection, and second reflection. Following the above, this ARM was compared to MRM, and the results are shown in Table 8. The proposed system adapt to an altered map by adapting the radio map database. The user create a new csv file based on the altered map as an input as shown in Figure 3. Then the system will generate a new radio map based on ray tracing propagation model using Matlab. At present, the creating of csv files is still manual; in the future, it can be done automatically using image processing.

MSE-RT is the mean squared error obtained using the ray-tracing model, while MSE-MW is the mean squared error using the multi-wall model. Table 8 and Figure 10 show the RSSI error. The MSE-RT was smaller than the MSE-MW, meaning that the ray-tracing model was better than the multi-wall model in predicting RSSI. This is because MW only counted the direct signal between TX and RX, while RT also calculated indirect signals such as reflection and diffraction. Therefore, the RSSI predictions using RT are more accurate than MW.

Some previous studies on ray-tracing propagation models have succeeded in obtaining results of MAE ranging from 3 to 8.52 [64,65,66,67]. The results of this study showed an average MAE of 2.9—A much better result than the results of the above studies. Other studies used RMSE to measure the error of the system. The average RMSE obtained in this study was 3.67, which is still better than some previous studies that obtained an RMSE above 4 [57,58,68,69]. The accuracy of the RSSI prediction based on the ray-tracing propagation modeling is, therefore, very high.

### 4.3. The Development of RSSI Adaptation

The results of the first experiment are presented in Figure 11. As a comparison, when no people are blocking the LOS, the values of RSSI received by the MD were −45 dBm for AP1 and AP2, and −46 dBm for AP3. It appears that when people bar the signal path between AP and MD and stand in the LOS path, the value of RSSI reduces, especially when the people’s positions are close to the MD.

The difference in the values of RSSI when there is one person as opposed to no persons (Δ RSSI) is shown in Figure 12. RSSI decreased by an average of 5 dBm when one person was 1 m to 3 m away from the MD. When the distance of the person was 4 m to 6 m, then, the RSSI decreased by an average of 3 dBm. The RSSI further decreased by 1 dBm when the distance of the person was 7 m to 9 m. An average value (5 dBm) was used as the value of the LOS PPE constant, later used in the RSSI adaptation for the development of AIRY.

Figure 12 shows that the closer the distance of people to the MD in the LOS position, the greater the decrease in RSSI. The closer the people are to the MD, the more the people will block the signal from the transmitter. This case applied to the RSSI originating from AP1, AP2, and AP3.

The results of the second experiment are shown in Figure 13. The RSSI values are still around 48 dBm, i.e., between 47 dBm and 49 dBm. As a reference, the RSSI when there was no one was 47.34 dBm for AP1, 48 dBm for AP2, and 48 dBm for AP3. It can be seen in Figure 11 that when one person was around the MD, at a varied position and distance, the value of RSSI did not significantly reduce compared to the RSSI obtained when there were no people.

The average difference in RSSI between the conditions in which one person was around the MD as opposed to no people can be seen in Table 9. Therefore, the further the distance between the people and the MD, the smaller the impact of the people on RSSI. People at a distance of 1 m from the MD reduced the RSSI of the MD by around 0.5 dB. This value then decreased to around 0.4 dB when the distance was 2 m and around 0.3 dB at 3 m. The maximum value (0.5 dBm) was, therefore, used as the value of the NLOS PPE constant, later used in the RSSI adaptation.

The decrease in RSSI, in this case, was much smaller than the decrease in RSSI when the people were at the LOS position. This is because the people in the LOS position blocked the direct signal and caused significant attenuation, whereas the people in the NLOS position mainly blocked the reflection signal. The reflection signal power was much smaller than the direct signal.

Both previous experiments discussed the effect of one person around MD on RSSI. The third experiment aimed to determine the effects of many people around the MD on RSSI, ranging from 2 people to 13 people, who had varied positions and distances from the MD—Starting at a distance of 1 m and then proceeding with distances of 2 m and 3 m.

The results of the experiment with a people distance of 1 m can be seen in Figure 14, which explains that the more people around the MD, the higher the decrease in RSSI on the MD; hence, the difference in RSSI of many people compared to no people also increased. The RSSI started to reduce by 0.5 dBm when 2 people were present, and then reached 0.9 dBm with 7 people. The more people around the MD, the more reflection signals were blocked, such that the RSSI value decreased even more.

The results of the experiment with people at a distance of 2 m are shown in Figure 15. The number of people ranged from 2 to 10 at varied positions, as shown in Table 6. The average value for the difference in RSSI was almost the same as the results of previous experiments, but the results of this experiment were smaller. The further the people position from the MD, the lesser the impact of the people on the decline in RSSI.

Figure 16 shows the average RSSI results of experiments according to the number of people involved. The results were generally the same as the previous experiment, but for the same number of people, the ΔRSSI was smaller than the results of previous experiments. This result shows that the people distance from the MD affected the RSSI received by the MD.

The three experiments above show that the number, the position, and the distance of people to the MD in the NLOS position affected the value of RSSI; the more the people, the greater the influence, and the closer the distance of the people to the MD, the greater the influence.

Finally, the rules of the RSSI adaptation are (see Figure 17 and Algorithm 2) listed below:If there are people in the NLOS position between the AP position and the initial position, then 0.5 dBm is added to the RSSI.If the people are in the LOS position between the AP position and the initial position, then 5 dBm is added to the RSSI.Then, if there are no people, the RSSI value does not change.

**Algorithm 2:** RSSI adaptation
**Input**:Initial RSSI,People Location, APs Location, Initial Position
**Output**: Adapted RSSI
**1:**

Get Initial RSSI (Ri)and People Location 

**2:**

Get Initial Location and APs Location

**3:**

For each Ri do

**4:**
  If (there are no people) then
**5:**
  Ra=Ri
**6:**
  End If
**7:**
  If (there are people in the LOS position) then
**8:**
  Ra=Ri+5
**9:**
  End If
**10:**
  If (there are people in the NLOS position) then
**11:**
  Ra=Ri+0.5
**12:**
  End If
**13:**

End For


### 4.4. The Development of an Accurate IPS Based on People Effect and Ray-Tracing (AIRY)

#### 4.4.1. The Accuracy of IPS without the Influence of People

Figure 18 shows the steps for checking IPS accuracy. The first step was to create a radio map database. There are 3 types of databases, as explained in the methodology section. The location used for testing was the 3rd floor of Razak Tower, Universiti Teknologi Malaysia Kuala Lumpur.

The RSSI vectors in 65 test points were recorded. At each test point, 30 RSSI data points from AP1, AP2, and AP3 were recorded. A total of 65 sample nodes were used to check the system accuracy. These sample nodes were measured in the online stage using MD. These data were then matched with the radio map databases using the KNN algorithm to obtain a location estimate for the MD/user.

The positioning algorithm used in this experiment is the KNN algorithm. This is because there is a previous research by Iyad that has same technique, same devices and same place (the 3rd floor of the UTM Kuala Lumpur Razak Tower) [48]. Iyad nominated 2 positioning algorithms: KNN and ANN. The results obtained showed that KNN can achieve more accurate positioning results than ANN. The KNN algorithm with k = 3 is chosen because it produces the highest percentage of accuracy. Another reason is because KNN is simpler than ANN. Furthermore, ANN is a complex system that needs heavy computations especially in the training phase, and this complexity does not fit with mobile device limitation.

The error values of the 65 test points for the ARM-RT database are shown in Figure 19. The maximum errors of the MRM database, the ARM-RT database, and the ARM-MW database were 1.37 m, 1.47 m, and 2 m, respectively.

The smallest average error (0.62 m) was obtained using the MRM database followed by ARM-RT and ARM-MW, with an average accuracy of 0.97 m and 1.27 m, respectively. The localization error using MRM database is significantly lower than in RADAR (a radio-frequency based system for locating and tracking users inside buildings) [41], by a factor of four (0.62 vs. 2.5 m) because AIRY used more radio map database than RADAR. RADAR collected RSSI vectors at 70 reference points (RPs) for the floor that has dimension of 43.5 m by 22.5 m, and AIRY has 523 RSSI vectors for the floor that has a dimension of 80 × 16 m. The number of RPs has a profound effect on the fingerprint positioning methods. Too few results in inaccurate fingerprint data, leading to poor performance

The best accuracy was achieved by the MRM database. However, the MRM database requires more time and energy to construct. The highest average error using the ARM-MW database was 1.27 m, similar to the results of previous studies conducted in the same building by Iyad [48]. A better average error, 0.97 m, was obtained when using the ARM-RT database. The time needed to create an ARM-RT database was indeed longer than for the ARM-MW database. The generation of ARM-RT took around 25 to 30 min while ARM-MW took only 5 to 10 min to generate. Based on error value, ARM-RT is the most promising choice of all the databases. The accuracy of the RSSI prediction greatly affects the accuracy of IPS. After testing the accuracy of IPS with 3 types of radio maps, the next step involved testing the accuracy of the system when PPE was accounted for.

#### 4.4.2. The Accuracy of IPS Considering the People Presence around the User

The more RSSI from APs affected by PPE, the less accurate the system. As a comparison, the accuracy of the system when there were no people was 0.62 m when using the MRM, 0.97 m when using ARM-RT, and 1.27 m when using ARM-MW. Figure 20 shows the accuracy of IPS when the effect of people around the user in the NLOS position was considered. The presence of people caused a decrease in accuracy of 33%, 44%, and 57% for the affected 1 AP, 2 APs, and 3 APs, respectively.

The influence of people present in the LOS position was greater than in the NLOS position. The presence of people in the LOS position caused a decrease in accuracy of 273%, 322%, and 334% for the affected 1 AP, 2 APs, and 3 APs, respectively, using the ARM-RT database.

The paper focuses on the influence of people around the user so that during data retrieval and testing the mobile device (MD) is not held by human experimenter but the mobile device is fixed on a tripod. Based on the experiments, the accuracy of the proposed model is not affected by the phone orientation. However, if the MD is held by a human experimenter, the MD orientation will affect the accuracy because the MD orientation affects the human experimenter’s position on MD and access points.

PPE had a significant effect on IPS accuracy. In the worst case, in which the RSSI from the three APs was influenced by people in the LOS, IPS accuracy was reduced by 334%. Hence, the ability of the proposed model (AIRY) to overcome this problem was tested to observe any evidence of improved accuracy.

#### 4.4.3. The Accuracy of the Proposed Model (AIRY)

The ability of the system to overcome the issue of the presence of people in this experiment was examined using 65 testing points. The system targeted accuracy improvements of less than 1 m. The average accuracy in the initial position was 2.04 m. After the RSSI adaptation process, which considered the people effect, a new average accuracy of 0.57 m was obtained, indicating a very significant improvement in accuracy. The proposed model succeeded at reaching the target of under 1 m accuracy.

The AIRY model requires the knowledge of people’s positions around the MD. Outside the controlled experiment, at this time this knowledge can be acquired manually by users. There is a feature in the AIRY apps that can be used by the user to provide the information about people’s positions around the MD. In future work, it can use image processing to provide this information automatically based on an image from closed-circuit television (CCTV). The proposed approach tracked the position of people with 3m distance from MD. Then based on people’s positions around MD, initial position of MD, and position of APs, the system will determine the position of the people whether in the LOS or NLOS position to apply the RSSI adjustments (RSSI adaptation).

A total of 3 RSSIs were recorded by the MD as emitted by 3 APs. The people influenced each RSSI when they were positioned around the MD in the LOS or NLOS positions. Because there were 3 APs, there were also 8 variations in the position of the people, as shown in Table 10. When the positions of the people against the 3 APs were all in the NLOS, an initial error value of 1.92 m was obtained. When the position of the people against 3 APs was all in the LOS, the initial error value increased to 2.11 m. This is because the influence of people in the LOS position on RSSI was greater than those in the NLOS position.

A comparison of the accuracy of the proposed system with that of related works is shown in Table 11. The proposed method using the ray-tracing propagation model and considering the people’s presence effect was able to achieve high accuracy (0.57 m). AIRY and Adaptive Indoor Positioning System (DIPS) both used radio propagation and the KNN algorithm. AIRY achieved better accuracy than DIPS, as AIRY used ray-tracing, which focuses on multipath signals and therefore significantly affecting the accuracy of RSSI predictions.

## 5. Conclusions

In this paper, an accurate indoor-positioning model based on WLAN fingerprinting was designed and validated. This model addressed the drawbacks of the traditional fingerprint method by generating an automatic RM database based on a ray-tracing model (ARM-RT), an indoor propagation model that considers the multipath effect, and an accurate indoor-positioning model based on the people effect and ray-tracing (AIRY). ARM-RT is the major contribution of this research. It is a solution to one of the main problems in IPS WLAN fingerprinting because RM construction requires a lot of time and energy. RM construction not only focuses on the direct signal from the AP to the MD but also takes into account reflection. The construction of a manual radio map for the 3rd floor of Razak Tower took 8 h, while the automatic radio map construction using ray-tracing only took about 30 min. Also, very satisfactory MSE values were obtained compared to the automatic radio map that used multi-wall models, where 12.19 ${\mathrm{dBm}}^{2}$ was obtained for the ray-tracing model and 92.38 ${\mathrm{dBm}}^{2}$ was recorded for the multi wall-model. The system accuracy was also very good, which was 0.97 m. The main contribution of this study is a highly accurate positioning system. Such high accuracy was obtained by adding an adaptation process in the initial RSSI. The adapted RSSI vectors were calculated based on initial RSSI as input and considering the presence of people around the initial position. The rules that applied to the adaptation of the RSSI vector were as follows: if there are people in the NLOS position (between the APx position and the initial position), the RSSI value of the APx received by MD was increased by 0.5 dBm (NLOS PPE constant), and if the people are in the LOS position, RSSI was increased by 5 dBm (LOS PPE constant). Then, if there are no people, the RSSI value does not change. The average accuracy in the initial position was 2.04 m. After the RSSI adaptation process, a new average accuracy of 0.57 m was obtained, indicating a very significant improvement in accuracy.

## Figures and Tables

**Figure 1 sensors-19-05546-f001:**
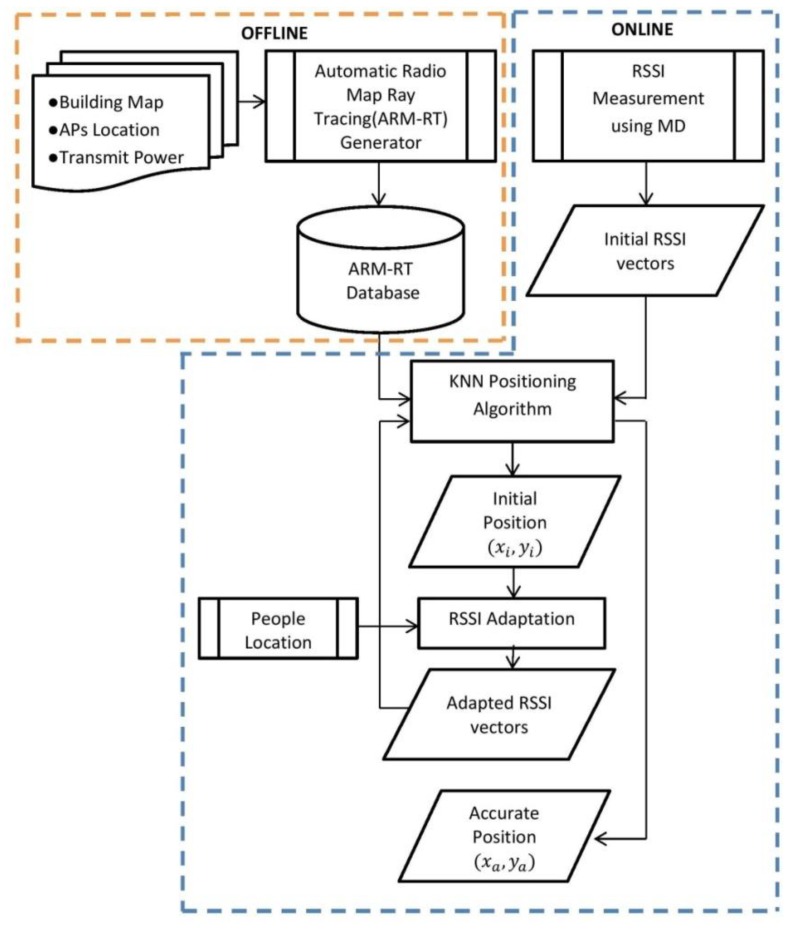
The proposed model for accurate indoor positioning considering the people effect and multipath signal propagation (AIRY).

**Figure 2 sensors-19-05546-f002:**
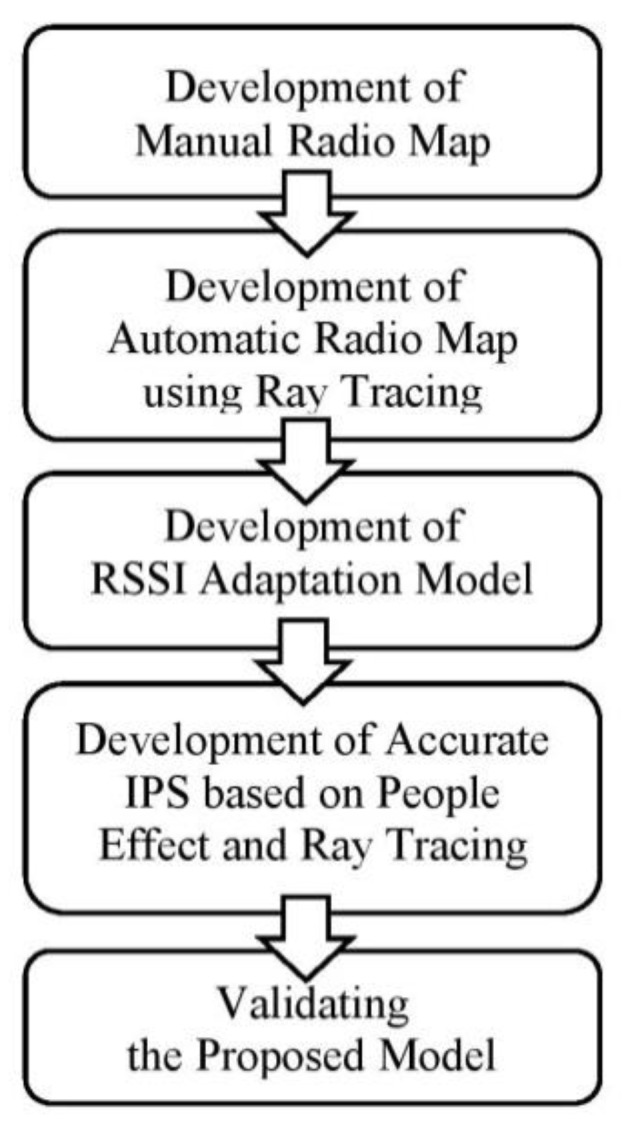
Method of research.

**Figure 3 sensors-19-05546-f003:**
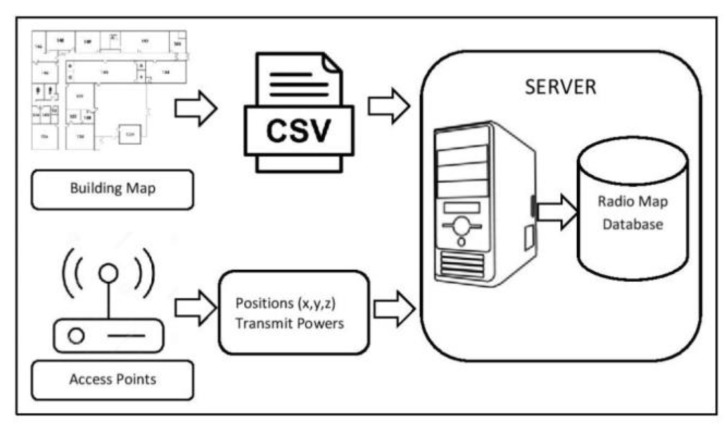
Process diagram of automatic radio map generator using ray-tracing (ARM-RT) development.

**Figure 4 sensors-19-05546-f004:**
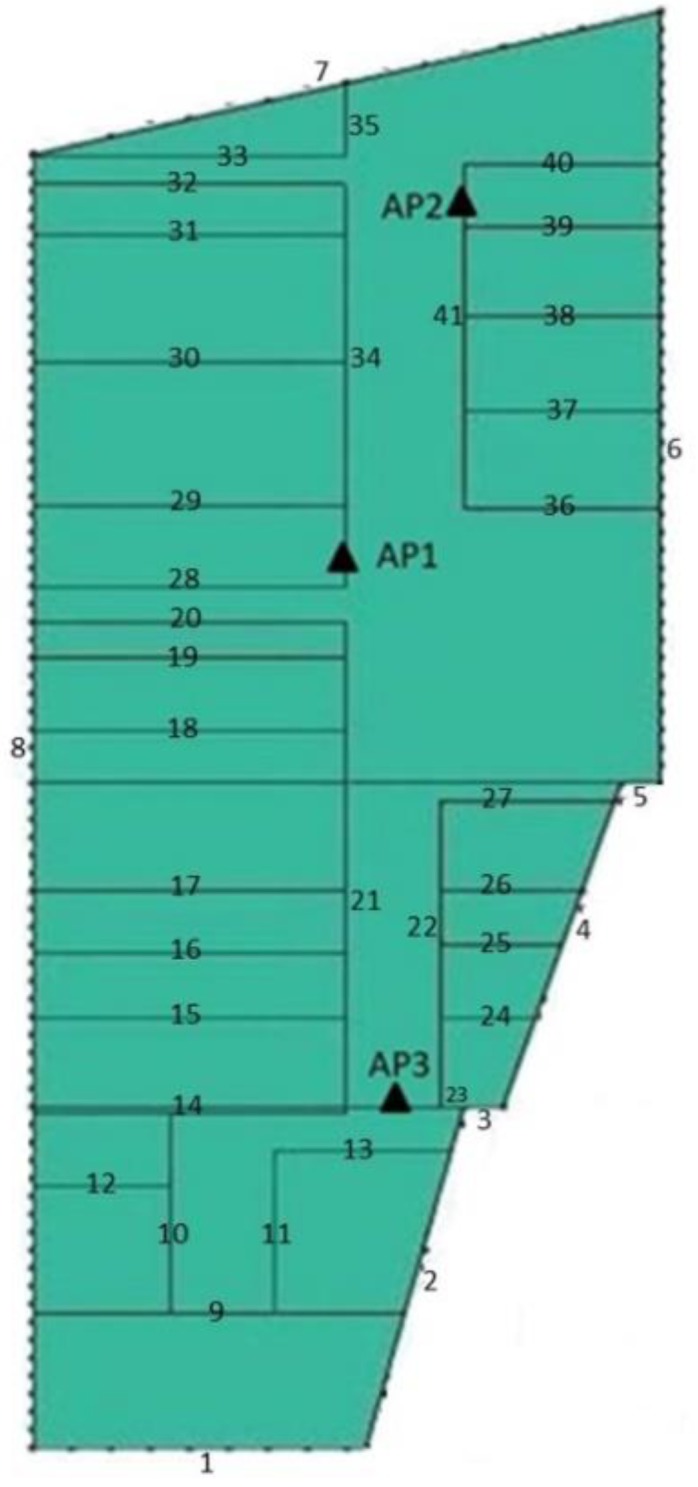
The layout of the walls on Razak Tower Level 3.

**Figure 5 sensors-19-05546-f005:**
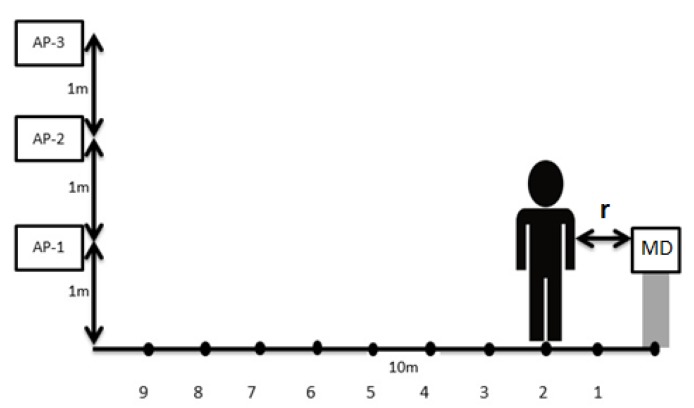
People in the line of sight (LOS) position between transmitter TX (access points, APs) and receiver RX (mobile device, MD).

**Figure 6 sensors-19-05546-f006:**
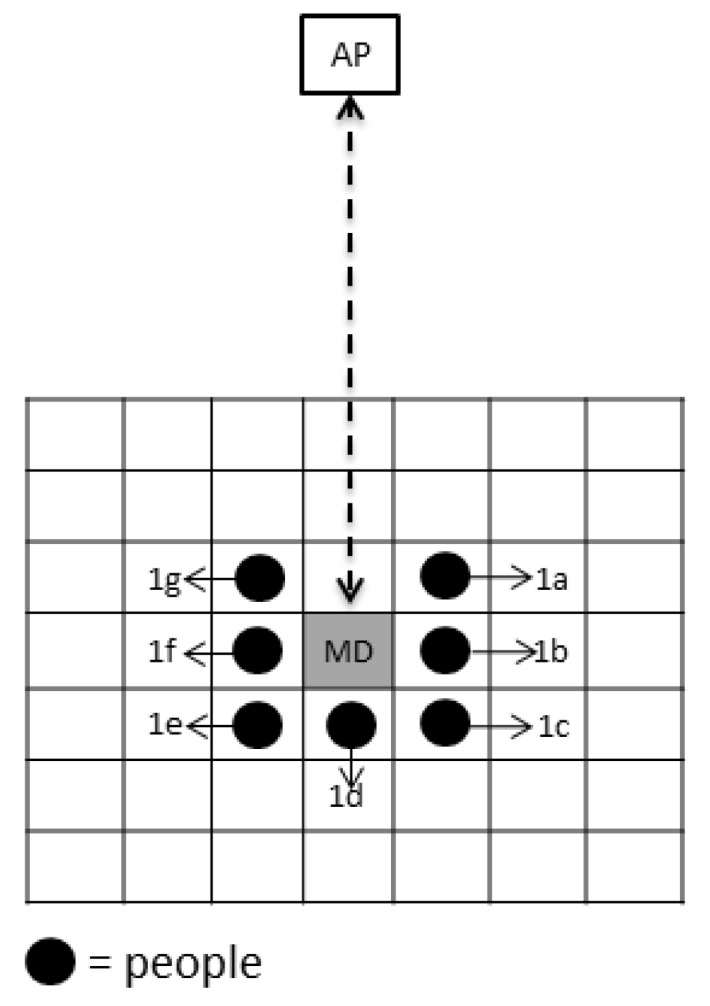
Top view of the people position around MD (the first ring).

**Figure 7 sensors-19-05546-f007:**
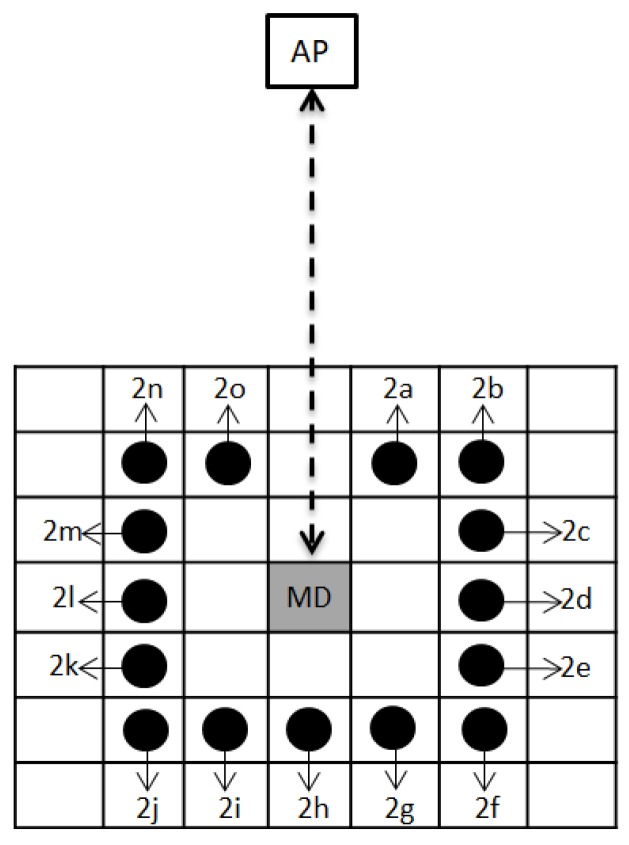
Top view of the people position around MD (the second ring).

**Figure 8 sensors-19-05546-f008:**
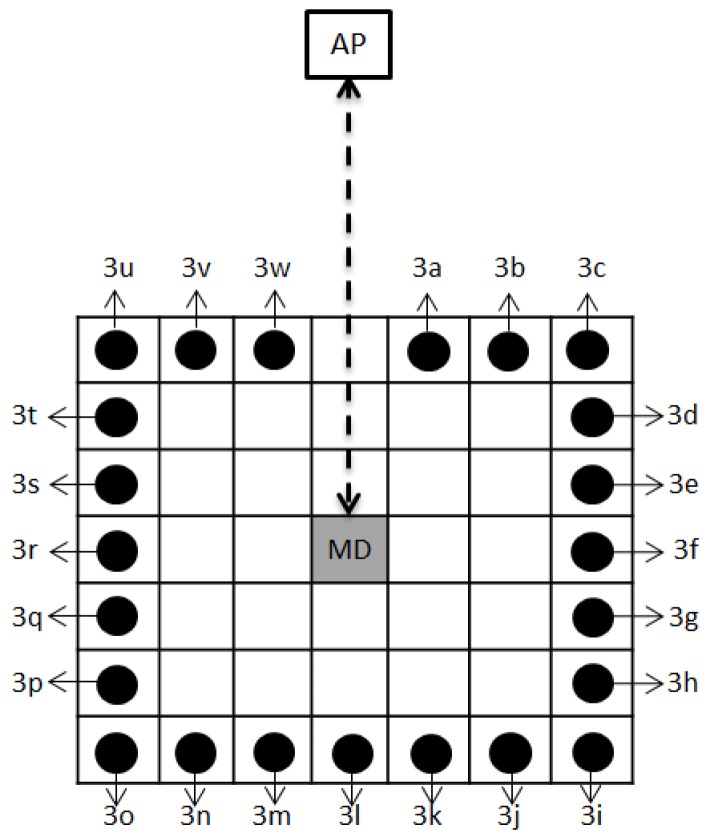
Top view of the people position around MD (the third ring).

**Figure 9 sensors-19-05546-f009:**
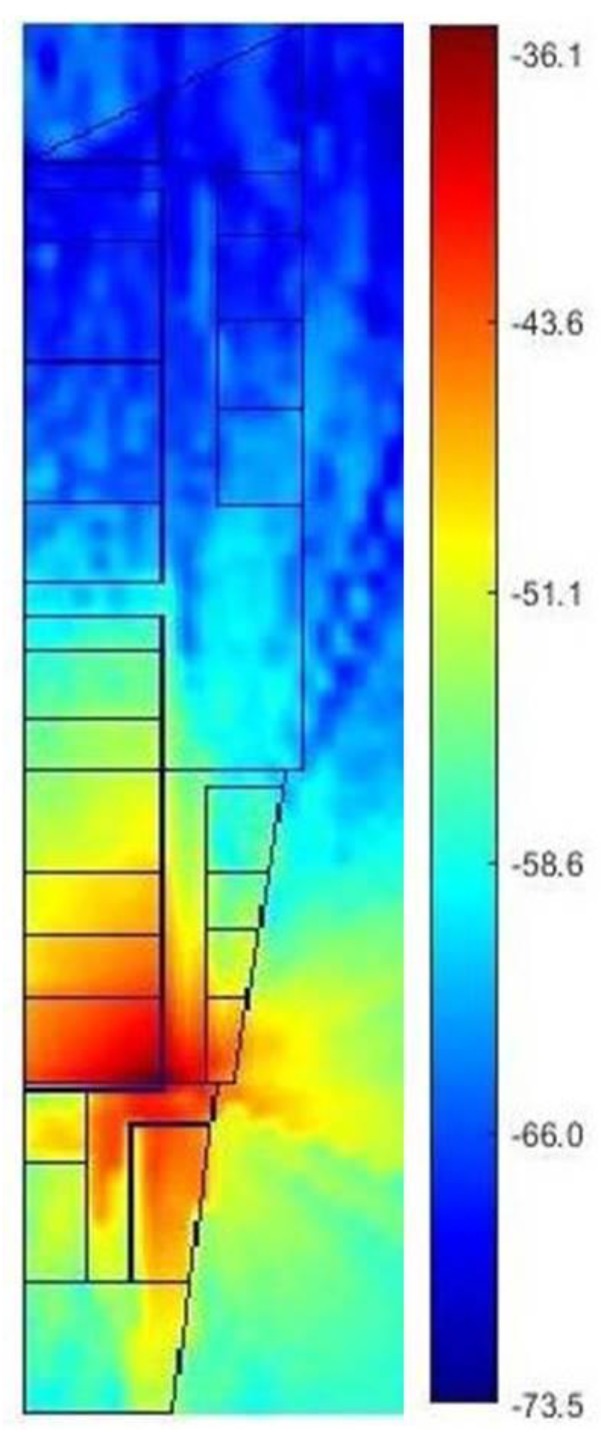
Visualization of received signal strength indicator (RSSI) prediction from AP3 on the third floor of Razak Building.

**Figure 10 sensors-19-05546-f010:**
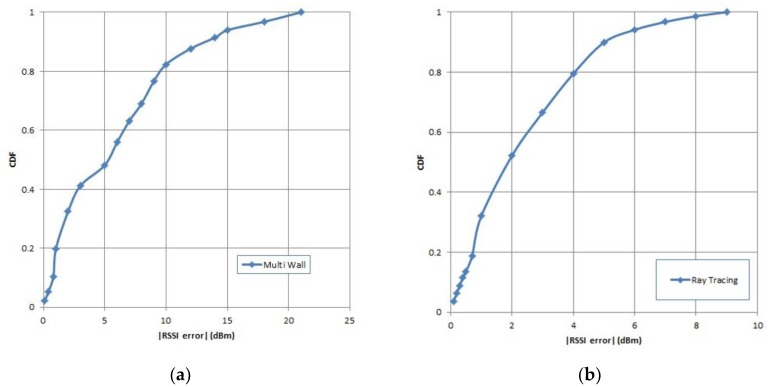
Cumulative distribution function (CDF) of RSSI error using (**a**) multi-wall (**b**) ray-tracing.

**Figure 11 sensors-19-05546-f011:**
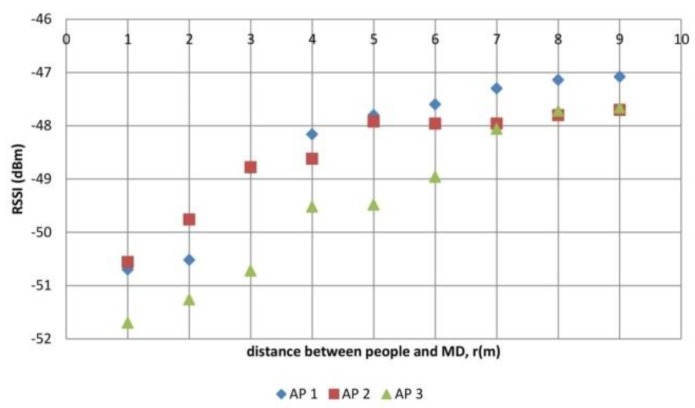
RSSI because of one person effect in LOS position.

**Figure 12 sensors-19-05546-f012:**
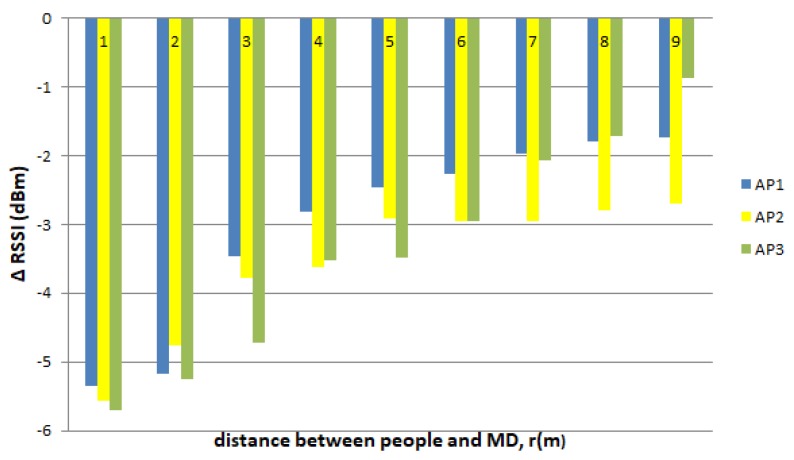
The difference in RSSI when there is one person in the LOS position as opposed to no one.

**Figure 13 sensors-19-05546-f013:**
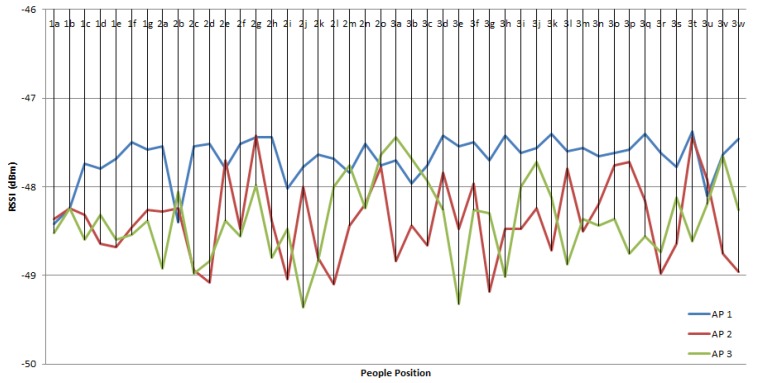
The effect of one person around the MD on RSSI.

**Figure 14 sensors-19-05546-f014:**
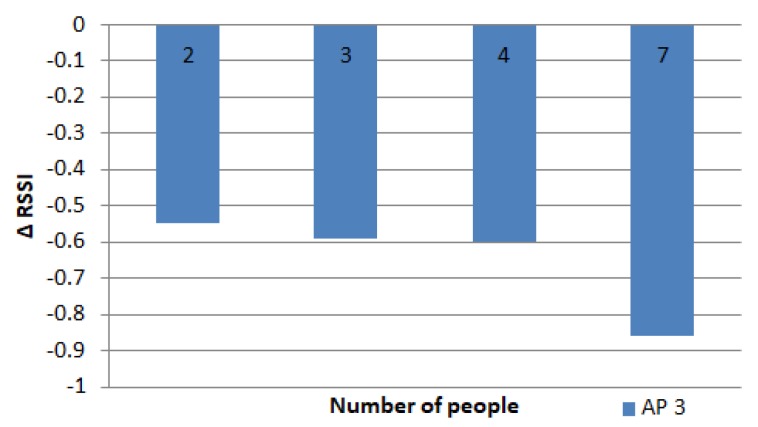
The effect of many people on the RSSI in the 1st ring.

**Figure 15 sensors-19-05546-f015:**
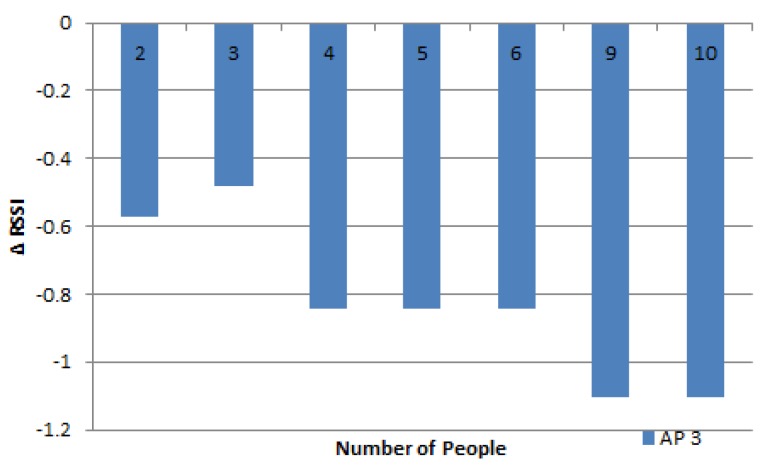
The effect of many people on RSSI in the 2nd ring.

**Figure 16 sensors-19-05546-f016:**
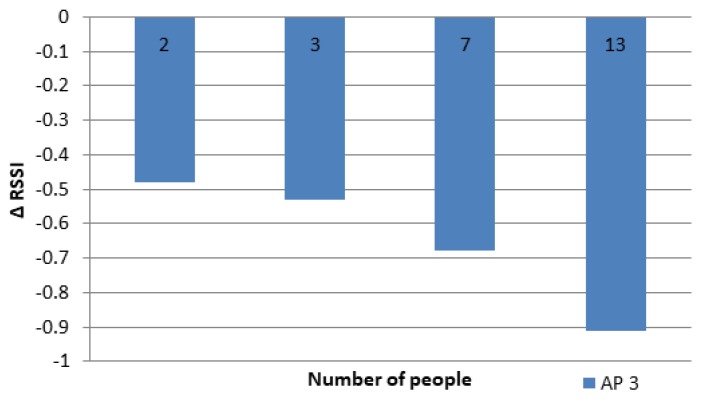
The effect of many people on the RSSI in the 3rd ring.

**Figure 17 sensors-19-05546-f017:**
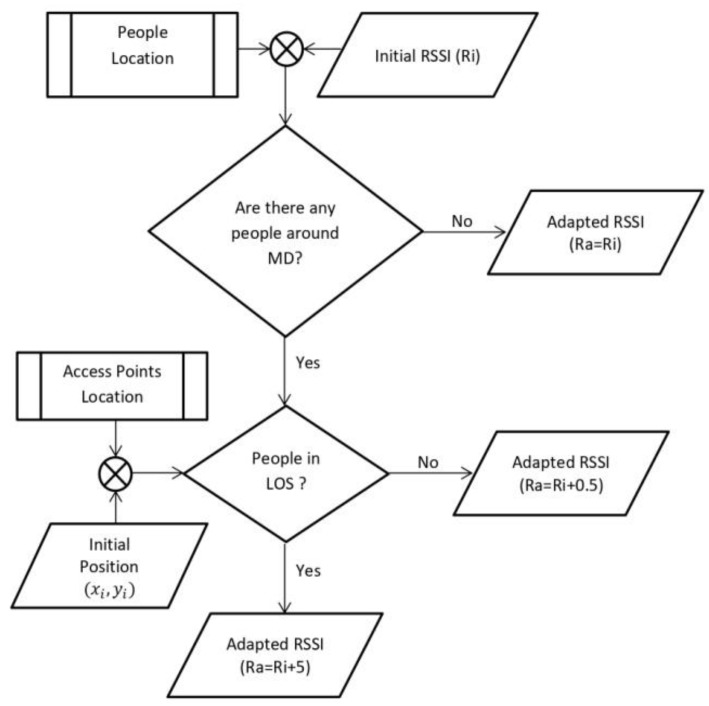
RSSI adaptation process.

**Figure 18 sensors-19-05546-f018:**
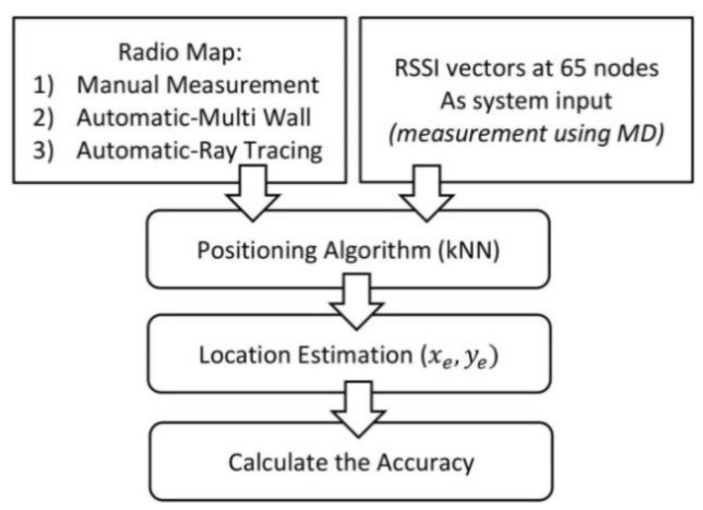
Steps to find accuracy.

**Figure 19 sensors-19-05546-f019:**
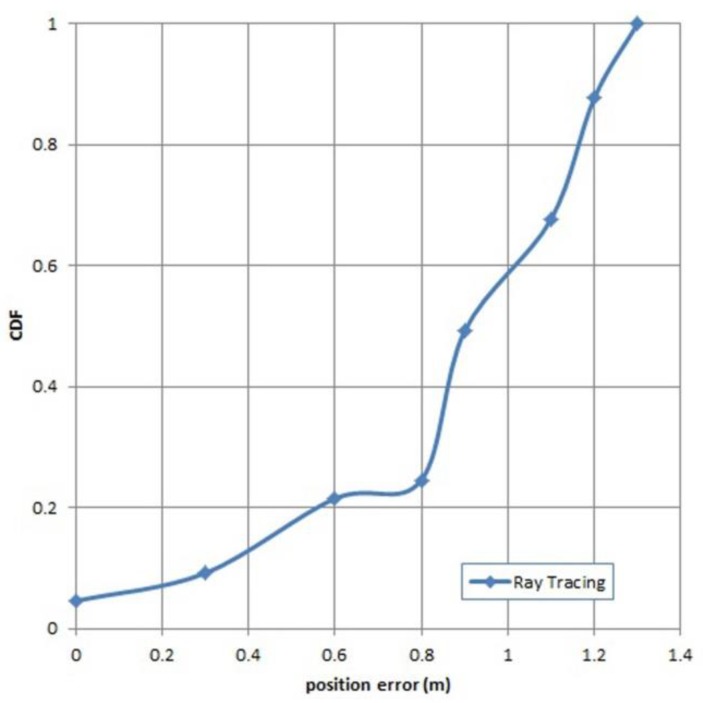
CDF of the the accuracy of the system using the ray-tracing automatic radio map.

**Figure 20 sensors-19-05546-f020:**
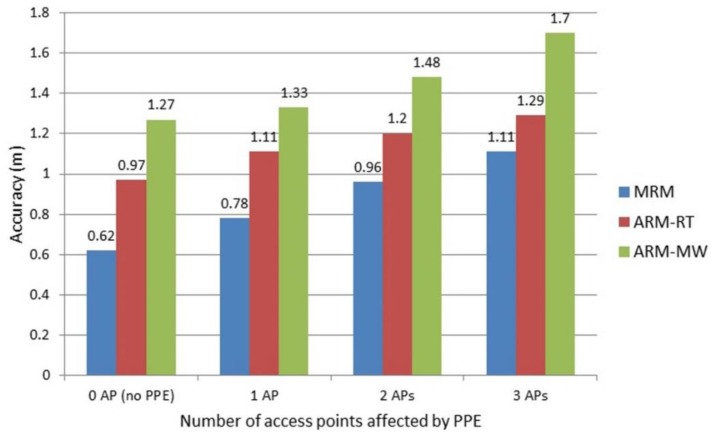
The average accuracy of indoor-positioning system (IPS) due to the influence of people present in the NLOS position.

**Table 1 sensors-19-05546-t001:** Material properties used for ray tracing [57].

Material	Relative Permittivity $\left(\varepsilon_{r}\right)$
Brick wall	5.86
Reinforced concrete	7.00
Wooden door	2.58
Window	6.38
Wooden floor	2.58
Ceiling	5.86

**Table 2 sensors-19-05546-t002:** The material of the walls on Level 3 of Razak Tower.

Walls	Material
1, 3, 5, 10, 12, 18–20, 27–28, 40	Brick
2, 4, 6–8, 22–23, 41	Glass
9, 11, 13–17, 21, 24–26, 29–39	Particleboard

**Table 3 sensors-19-05546-t003:** Parameters of the ray-tracing model simulation.

Parameters	Value
Frequency	2.4 GHz
Method	Image Method
Multipath	Transmission, 1st and 2nd Reflections
Transmit power	−30 dBm

**Table 4 sensors-19-05546-t004:** Parameters of the ray-tracing simulation.

Parameter	Value
Number of access points (APs)	3
Height of APs	2.9 m
Relative Permittivity of glass	6.38
Relative Permittivity of the brick wall	5.86
Relative Permittivity of the particleboard	2.70
Number of walls	41
Number of rooms	20

**Table 5 sensors-19-05546-t005:** Position variation of many people around MD in the first ring (see Figure 6).

People Position in the First Ring				
2 people		3 people	4 people	6 people
• 1b, 1f • 1a, 1g • 1c, 1e • 1b, 1d	• 1d, 1f • 1c, 1g • 1a, 1e	• 1a, 1d, 1g • 1b, 1d, 1f • 1a, 1b, 1c • 1c, 1d, 1e • 1e, 1f, 1g	• 1a, 1c, 1e, 1g • 1a, 1b, 1f, 1g • 1b, 1c, 1e, 1f	• 1a, 1b, 1c, 1e, 1f, 1g **7 people** • 1a, 1b, 1c, 1d, 1e, 1f, 1g

**Table 6 sensors-19-05546-t006:** Position variation of many people around MD in the second ring (see Figure 7).

People Position in the Second Ring			
2 people	3 people	5 people	9 people
• 2d, 2l • 2b, 2n • 2f, 2j • 2d, 2h • 2h, 2l • 2f, 2n • 2b, 2j	• 2b, 2h, 2n • 2d, 2h, 2l • 2c, 2d, 2e • 2g, 2h, 2i • 2k, 2l, 2m **4 people** • 2b, 2f, 2j, 2n	• 2b, 2c, 2d, 2e, 2f • 2j, 2k, 2l, 2m, 2n • 2f, 2g, 2h, 2i, 2j **6 people** • 2b, 2c, 2d, 2l, 2m, 2n • 2d, 2e, 2f, 2j, 2k, 2l	• 2b, 2c, 2d, 2e, 2f, 2g, 2h, 2i, 2j • 2f, 2g, 2h, 2i, 2j, 2k, 2l, 2m, 2n **10 people** • 2b, 2c, 2d, 2e, 2f, 2j, 2k, 2l, 2m, 2n

**Table 7 sensors-19-05546-t007:** Position variation of many people around MD in the third ring (see Figure 8).

People Position in the Third Ring				
2 people		3 people	4 people	13 people
• 3f, 3r • 3c, 3u • 3i, 3o • 3f, 3l • 3l, 3r	• 3i, 3u • 3c, 3o	• 3c, 3l, 3u • 3f, 3l, 3r • 3e, 3f, 3g • 3k, 3l, 3m • 3q, 3r, 3s	• 3c, 3i, 3o, 3u **7 people** • 3c, 3d, 3e, 3f, 3g, 3h, 3i • 3i, 3j, 3k, 3l, 3m, 3n, 3o • 3o, 3p, 3q, 3r, 3s, 3t, 3u	• 3c to 3o • 3i to 3u

**Table 8 sensors-19-05546-t008:** Root mean squared error (MSE) and mean absolute error (MAE) of ray-tracing and multi wall.

RMSE		MAE	
RT	MW	RT	MW
3.7	8.6	2.9	6.3

**Table 9 sensors-19-05546-t009:** The average difference in RSSI for the conditions of one people around MD versus no people.

People Position	Δ RSSI		
	AP1	AP2	AP3
1st Ring	−0.51	−0.42	−0.45
2nd Ring	−0.35	−0.39	−0.38
3rd Ring	−0.27	−0.35	−0.30

**Table 10 sensors-19-05546-t010:** Variation of people position from AP and MD.

No.	People Position from AP and MD		
	AP1	AP2	AP3
1	NLOS	NLOS	NLOS
2	NLOS	NLOS	LOS
3	NLOS	LOS	NLOS
4	NLOS	LOS	LOS
5	LOS	NLOS	NLOS
6	LOS	NLOS	LOS
7	LOS	LOS	NLOS
8	LOS	LOS	LOS

**Table 11 sensors-19-05546-t011:** Comparison of Accuracy with Related Works.

Research	Method Used	Accuracy
Bahl [41]	Manual Radio Map + KNN	2.5 m
Hung-Huan [70]	Path Loss model + Triangulation	1.6 m
Vahidnia [71]	Manual Radio Map + BPM	1.4 m
Sun [72]	Manual Radio Map + WKNN	1.2 m
DIPS [48]	Path Loss Model + KNN	1.2 m
AIRY	Ray-tracing + KNN	0.57 m

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
