# Peer review of "Accurate Indoor-Positioning Model Based on People Effect and Ray-Tracing Propagation"

_sensors, 2019, doi:10.3390/s19245546_

Round 1
Reviewer 1 Report
The paper is overall well-written and easy to read. An important problem is investigated regarding the human presence effect.
The concerns are as follows:
The ray tracing relies on the availability of the indoor map. How does the proposed system adapt to an altered map? Does the phone orientation affect the indoor localization performance?
Author Response
Point 1: The ray tracing relies on the availability of the indoor map. How does the proposed system adapt to an altered map?
Response 1: The proposed system adapt to an altered map by adapting the radio map database. The user creates a new csv file based on the altered map as an input as shown in figure 3. Then the system will generate a new radio map based on ray tracing propagation model using Matlab. At present, the creating of csv files is still manual, in the future it can be developed automatically using image processing.
Point 2: Does the phone orientation affect indoor localization performance?
Response 2: As stated in 4.4.2, this paper focuses on the influence of people around the user so that during data retrieval and testing the mobile device (MD) is not held by human experimenter but the mobile device is fixed on a tripod. Based on the experiments, the accuracy of the proposed model is not affected by the phone orientation. However, if the MD is held by a human experimenter, the MD orientation will affect the accuracy because the MD orientation affects the human experimenter's position on MD and access points.

Reviewer 2 Report
This paper does not describe the type of approach, i.e. device-aided (localizing e.g. a smartphone with active wifi card) or device free (using fingerprinting techniques). Furthermore, the device-aided approach may be carried out at the device-side or at the access-point-side. It seems the paper follows a device-aided approach, but the introduction section should discuss also the device-free approach which has a more general application, e.g. make reference to:
D. Konings, F. Alam, F. Noble and E. M. Lai, "Device-Free Localization Systems Utilizing Wireless RSSI: A Comparative Practical Investigation," in IEEE Sensors Journal, vol. 19, no. 7, pp. 2747-2757, 1 April1, 2019.
G. Pecoraro, S. Di Domenico, E. Cianca, M. De Sanctis, "LTE Signal Fingerprinting Localization based on CSI", 2017 IEEE 13th International Conference on Wireless and Mobile Computing, Networking and Communications, WiMob 2017, Rome (Italy), 9-11 October, 2017.
Please, discuss more deeply this theme.
Author Response
Point 1: This paper does not describe the type of approach, i.e. device-aided (localizing e.g. a smartphone with active wifi card) or device free (using fingerprinting techniques). Furthermore, the device-aided approach may be carried out at the device-side or at the access-point-side. It seems the paper follows a device-aided approach, but the introduction section should discuss also the device-free approach which has a more general application, e.g. make reference to:
Konings, F. Alam, F. Noble and E. M. Lai, "Device-Free Localization Systems Utilizing Wireless RSSI: A Comparative Practical Investigation," in IEEE Sensors Journal, vol. 19, no. 7, pp. 2747-2757, 1 April1, 2019. Pecoraro, S. Di Domenico, E. Cianca, M. De Sanctis, "LTE Signal Fingerprinting Localization based on CSI", 2017 IEEE 13th International Conference on Wireless and Mobile Computing, Networking and Communications, WiMob 2017, Rome (Italy), 9-11 October, 2017.
Please, discuss more deeply this theme.
Response 1: The discussion about this theme will be added in the introduction section.
Indoor positioning can be classified as device-based and device-free. On device-based systems, users need a device to know their position, such as smartphone-based and tag-based indoor positioning [1]. Instead of a device-free system, the user does not need a device to know his position. Users here can be people or objects. Device-free localization based on signal strength (RSSI) has three main techniques: fingerprinting, link-based, and radio tomographic imaging. From the user side, device-free is certainly more practical. However, from the system side, it will be more complex. For example, it takes 6 to 20 transceiver nodes for fingerprinting techniques. Whereas with the same technique device-based only requires 3 transmitters. In fact, the number of transmitters (APs) available in a building is also limited.
Fingerprinting techniques can be applied to device-based and device-free systems. On device-free systems, a number of nodes (transceivers) will be installed on the building. Then each node will record the RSSI emitted by other nodes and forward the data to the computer for the positioning process. On device-based systems, the device held by the user will record RSSI from several transmitters in the building and can use the device to determine its position. WLAN fingerprinting can work either based on measurements of the Received Signal Strength Indicator (RSSI) or of the Channel State Information (CSI). CSI-based signal fingerprinting provides better accuracy [2]. However, among many devices available on the market, CSI is only available on a few devices using modified drivers.
Another thing to consider is that if there are many users whose locations will be detected, the device-free system will find it difficult to identify each user and his position. On the contrary, it is quite easy for device-based. The fact is that almost everyone has a smartphone now, so it's easy to apply device-based localization.

Reviewer 3 Report
The paper addresses two important problems in fingerprint-based indoor positioning: reducing the training efforts and reducing the effect of human presence on the fingerprints. For the first problem, the author apply the well-known ray-tracing simulation approach. The human presence, in turn, is accounted for by increasing the RSSI by a predefined constant. The proposed algorithms are validated with real-life experimental measurements. The paper is reasonably structured and is readable. Unfortunately, a number of shortcomings make the paper not ready for publication in its current form.
First of all, the experiment description lacks several important details, in particular about ground truth setup. The paper states "Data retrieval is done when there are no people around MD." What was the measurement procedure exactly? Was the mobile device fixed on a tripod in all experiments? Was it held by a human experimenter? Did the experimenter walked away when measuring each point? (If so, how far?)
Second, the AIRY model seems to require the knowledge of people's positions around the mobile device. How is this knowledge supposed to be acquired outside the controlled experiment? Does the proposed approach assume the position of all people in the environment is tracked? If it is tracked by AIRY, how would the system apply the RSSI adjustments? Do the positions converge to a stable results?
Third, the manual fingerprinting setup is very similar to that of RADAR [37]. However, the localization error reported in Section 4.4.1 is significantly lower than in RADAR, by a factor of four (0.62 vs 2.5 m). How can the authors explain the 4-times improvement of the localization accuracy using the same algorithm (kNN), in a similar testbed (building floor), and with only 3 APs (so it seems)?
The localization was done using kNN with k=3 and Euclidean distance metric. Why so? Did you consider other algorithms / parameters / metrics?
What was the used Wi-Fi standard / frequency band? The APs used in the experiments (Linksys WAP300N) seem to be capable of beam-forming, which could have significantly affected the experimental results.
Section 4 (experimental results) always refers to three APs. Where there only 3 APs on the whole floor (80x16 m)? Or only 3 APs were selected from those available (why? how?)
Table 9 has several issues:
- What is the meaning of the first three rows? Fingerprinting is normally done using several APs, so per-AP errors don't make much sense to this reviewer.
- The table mixes metrics with different units (meters and meters squared). These units should be labelled explicitly.
- The MSE metric should be removed, it simply duplicates the much more intuitive RMSE values.
In general, positioning performance in indoor positioning is best represented by CDF graphs (cumulative distribution function) of the error (see Section 3.B in [57]). Specifically, Table 9 and Figure 20 would be more informative as CDFs.
Figure 21 shows the accuracy with human presence; it would be useful to put the human-free results on the same figure for comparison.
Minor comments:
- Table 8 is redundant and can be removed.
- What is the resolution (grid step) of the automatically build radio map in Figure 10?
- The excessive amount of acronyms makes the paper harder to follow.
- Error values in the conclusion are missing the measurement units.
English/typos:
- The reviewer is not a native speaker, but the paper needs additional proofreading.
- "it does not needs"
- "triangulation, and fingerprint" -> "fingerprinting"
- "this model derived" -> "is derived"
- "properties of the materials are influence" -> remove "are"
- "compare to MRM by calculate" -> "by calculating"
- "Because of their intrinsic capability to simulate multipath propagation [55]." -> incomplete sentence
- "that will explain in detail"
- "one people" -> "one person"
- "Fisrt Ring" -> "First Ring"
- "Result and Discussion" -> "Results and Discussion"
- "when there are no people blocks LOS" -> "blocking"
Author Response
Point 1 : First of all, the experiment description lacks several important details, in particular about ground truth setup. The paper states "Data retrieval is done when there are no people around MD." What was the measurement procedure exactly? Was the mobile device fixed on a tripod in all experiments? Was it held by a human experimenter? Did the experimenter walked away when measuring each point? (If so, how far?)
Response 1 : The RSSI measurement procedure are the mobile device fixed on a tripod in all experiments. MD height is 1m above the floor. Then the experimenter walked away about 10m from MD when measuring each point. The experimenter pressed the button on the android application used to record RSSI. There is a time delay feature in the application before starting the RSSI recording so that there is time for the experimenter to walk away from MD. After the RSSI recording is finished there is a beep sound so the experimenter knows when he can return to MD.
Point 2 : Second, the AIRY model seems to require the knowledge of people's positions around the mobile device. How is this knowledge supposed to be acquired outside the controlled experiment? Does the proposed approach assume the position of all people in the environment is tracked? If it is tracked by AIRY, how would the system apply the RSSI adjustments? Do the positions converge to a stable results ?
Response 2 : The AIRY model requires the knowledge of people's positions around the MD. Outside the controlled experiment, at this time this knowledge can be acquired manually by users. There is a feature in the AIRY apps that can be used by user to provide the information about people’s positions around the MD. In the future work, it can use image processing to provide this information automatically based on image from CCTV. The proposed approach tracked the position of people with 3m distance from MD. Then based on people's positions around MD, initial position of MD, and position of APs, the system will determine the position of the people whether in the LOS or NLOS position to apply the RSSI adjustments (RSSI adaptation).
Point 3 : Third, the manual fingerprinting setup is very similar to that of RADAR [37]. However, the localization error reported in Section 4.4.1 is significantly lower than in RADAR, by a factor of four (0.62 vs 2.5 m). How can the authors explain the 4-times improvement of the localization accuracy using the same algorithm (kNN), in a similar testbed (building floor), and with only 3 APs (so it seems)?
Response 3 : The localization error reported in Section 4.4.1 is significantly lower than in RADAR, by a factor of four (0.62 vs 2.5 m) because AIRY used more radio map database than RADAR. RADAR collected RSSI vectors at 70 reference points (RPs) for the floor that has dimension of 43.5 m by 22.5 m, and AIRY has 523 RSSI vectors for the floor that has dimension of 80 x 16m. The number of RPs has a profound effect on the fingerprint positioning methods. Too few results in inaccurate fingerprint data, leading to poor performance (Xia et al. 2017).
Xia, Shixiong, Yi Liu, Guan Yuan, Mingjun Zhu, and Zhaohui Wang. 2017. “Indoor Fingerprint Positioning Based on Wi-Fi: An Overview.” ISPRS International Journal of Geo-Information 6 (5): 135.
Point 4 : The localization was done using kNN with k=3 and Euclidean distance metric. Why so? Did you consider other algorithms / parameters / metrics?
Response 4 : The positioning algorithm used in this experiment is the KNN algorithm. This is because there is a previous research by Alshami that has same technique, same devices and same place (the 3rd floor of the UTM Kuala Lumpur Razak Tower) [45]. Alshami nominated 2 positioning algorithms: KNN and ANN. The obtained results showed that KNN can achieve more accurate positioning results than ANN. The KNN algorithm with k = 3 is chosen because it produces the highest percentage of accuracy as shown in Figure 8 and Figure 9[45]. Another reason is because KNN is simpler than ANN. Furthermore, ANN is a complex system that needs heavy computations especially in the training phase, and this complexity does not fit with mobile device limitation.
KNN positioning result accuracy [45] |
ANN positioning result accuracy [45] |
Point 5 : What was the used Wi-Fi standard / frequency band? The APs used in the experiments (Linksys WAP300N) seem to be capable of beam-forming, which could have significantly affected the experimental results.
Response 5 : The APs (Linksys WAP300N) in this experiment is setting as Wi-Fi Access Point (default) operation mode and 2.4GHz frequency band. There is no beam-forming configuration here.
Point 6 : Section 4 (experimental results) always refers to three APs. Where there only 3 APs on the whole floor (80x16 m)? Or only 3 APs were selected from those available (why? how?)
Response 6 : There are only 3 APs on the whole floor (80x16m) of level 3 Menara Razak, but the detected Wi-Fi signal on the 3rd floor can be up to 6 SSID that come from other floors. The system was selected 3 APs based on their MAC address.
Point 7 : Table 9 has several issues:
- What is the meaning of the first three rows? Fingerprinting is normally done using several APs, so per-AP errors don't make much sense to this reviewer.
- The table mixes metrics with different units (meters and meters squared). These units should be labelled explicitly.
- The MSE metric should be removed, it simply duplicates the much more intuitive RMSE values.
Response 7 : The meaning of the first three rows is the RSSI estimation error (compare the RSSI estimation to real measurement of RSSI) for each AP (AP1, AP2, and AP3). Then based on the suggestion, Table 9 will only display the average error of the entire AP, no longer per AP. The error parameters displayed are RMSE and MAE.
RMSE |
MAE |
||
RT |
MW |
RT |
MW |
3.7 |
8.6 |
2.9 |
6.3 |
Point 8 : In general, positioning performance in indoor positioning is best represented by CDF graphs (cumulative distribution function) of the error (see Section 3.B in [57]). Specifically, Table 9 and Figure 20 would be more informative as CDFs.
Response 8 : We have constructed CDF graphs for Table 9 and Figure 20
Table 9 à CDF of RSSI error (Multi Wall) |
Table 9 à CDF of RSSI error (Ray Tracing) |
Figure 20 à CDF of position error (Ray Tracing) |
Point 9 : Figure 21 shows the accuracy with human presence; it would be useful to put the human-free results on the same figure for comparison.
Response 9 : The human-free results has been added on the same figure (Figure 21) for comparison.
Point 10 : Minor comments:
- Table 8 is redundant and can be removed.
- What is the resolution (grid step) of the automatically build radio map in Figure 10?
- The excessive amount of acronyms makes the paper harder to follow.
- Error values in the conclusion are missing the measurement units.
Response 10 :
-Table 8 is removed from the paper
-The resolution of the automatically build radio map in Figure 10 is 1m.
-The measurement units have been added
Point 11 : English/typos:
- The reviewer is not a native speaker, but the paper needs additional proofreading.
- "it does not needs"
- "triangulation, and fingerprint" -> "fingerprinting"
- "this model derived" -> "is derived"
- "properties of the materials are influence" -> remove "are"
- "compare to MRM by calculate" -> "by calculating"
- "Because of their intrinsic capability to simulate multipath propagation [55]." -> incomplete sentence
- "that will explain in detail"
- "one people" -> "one person"
- "Fisrt Ring" -> "First Ring"
- "Result and Discussion" -> "Results and Discussion"
- "when there are no people blocks LOS" -> "blocking"
Response 11: We fixed it and this paper is now in the proofread process. Thanks
Round 2
Reviewer 1 Report
The revision has addressed the concerns.
Multiple important references are missing.
1. He, S., & Shin, K. G. (2018, April). Steering crowdsourced signal map construction via Bayesian compressive sensing. In IEEE INFOCOM 2018-IEEE Conference on Computer Communications (pp. 1016-1024). IEEE.
2. Ferris, B., Fox, D., & Lawrence, N. D. (2007, January). Wifi-slam using gaussian process latent variable models. In IJCAI (Vol. 7, No. 1, pp. 2480-2485).
3. Shen, G., Chen, Z., Zhang, P., Moscibroda, T., & Zhang, Y. (2013). Walkie-Markie: Indoor pathway mapping made easy. In Presented as part of the 10th {USENIX} Symposium on Networked Systems Design and Implementation ({NSDI} 13) (pp. 85-98).
Author Response
Q1 : The revision has addressed the concerns.
A1 : Thank you very much
Q2: Multiple important references are missing.
He, S., & Shin, K. G. (2018, April). Steering crowdsourced signal map construction via Bayesian compressive sensing. In IEEE INFOCOM 2018-IEEE Conference on Computer Communications(pp. 1016-1024). IEEE. à [37] Ferris, B., Fox, D., & Lawrence, N. D. (2007, January). Wifi-slam using gaussian process latent variable models. In IJCAI(Vol. 7, No. 1, pp. 2480-2485). à [34]
Shen, G., Chen, Z., Zhang, P., Moscibroda, T., & Zhang, Y. (2013). Walkie-Markie: Indoor pathway mapping made easy. In Presented as part of the 10th {USENIX} Symposium on Networked Systems Design and Implementation ({NSDI} 13)(pp. 85-98). à [35]
A2: We have added these 3 references to the paper [34-35],[37].
Reviewer 2 Report
no additional comments
Author Response
Q1: no additional comments
A1: Thank you very much, and we have improved english in paper.